# A-to-I miR-378a-3p editing can prevent melanoma progression via regulation of *PARVA* expression

Guermarie Velazquez-Torres[1], Einav Shoshan[1], Cristina Ivan [2], Li Huang[1], Enrique Fuentes-Mattei [3], Harrison Paret[1], Sun Jin Kim[1], Cristian Rodriguez-Aguayo [3], Victoria Xie[4], Denise Brooks[5], Steven J.M. Jones[5], A. Gordon Robertson[5], George Calin[3], Gabriel Lopez-Berenstein[3], Anil Sood[2] & Menashe Bar-Eli[1]

Previously we have reported that metastatic melanoma cell lines and tumor specimens have reduced expression of ADAR1 and consequently are impaired in their ability to perform A-to-I microRNA (miRNA) editing. The effects of A-to-I miRNAs editing on melanoma growth and metastasis are yet to be determined. Here we report that miR-378a–3p is undergoing A-to-I editing only in the non-metastatic but not in metastatic melanoma cells. The function of the edited form is different from its wild-type counterpart. The edited form of miR-378a-3p preferentially binds to the 3′-UTR of the *PARVA* oncogene and inhibits its expression, thus preventing the progression of melanoma towards the malignant phenotype. Indeed, edited miR-378a-3p but not its WT form inhibits melanoma metastasis in vivo. These results further emphasize the role of RNA editing in melanoma progression.

[1] Department of Cancer Biology, Unit 1906, The University of Texas MD Anderson Cancer Center, 1515 Holcombe Blvd, Houston, TX 77030, USA.
[2] Department of Gynecologic Oncology, Unit 1362, The University of Texas MD Anderson Cancer Center, 1515 Holcombe Blvd., Houston, TX 77030, USA.
[3] Department of Experimental Therapeutics, Unit 1950, The University of Texas MD Anderson Cancer Center, 1515 Holcombe Blvd, Houston, TX 77030, USA. [4] Department of Neurosurgery, The University of Texas MD Anderson Cancer Center, 1515 Holcombe Blvd., Houston, TX 77030, USA. [5] Canada's Michael Smith Cancer Agency, Vancouver, BC V5Z4S6, Canada. Correspondence and requests for materials should be addressed to M.B-E. (email: mbareli@mdanderson.org)

Melanoma is the most aggressive type of skin cancer with an estimated 87,000 annual new cases in the United States, and close to 9,700 will result in death mostly due to metastasis[1]. Previously we have identified the CREB transcription factor as a "master switch" in melanoma metastasis by regulating genes involved in survival, angiogenesis and invasion[2–5]. CREB also regulates the expression of other important transcription factor involved in melanoma progression such as AP-2α and MITF[6,7]. Recently, we reported that CREB negatively regulates the expression of the ADAR1 enzyme, which is involved in A-to-I RNA editing of mRNAs and microRNAS (miRNAs)[8,9]. Indeed, we reported that metastatic melanoma cell lines and tumor specimens have reduced expression of ADAR1 and consequently are deficient in their ability to perform miRNAs A-to-I editing. We identified three miRNAs (miR-455-5p, miR-324-5p and miR-378a-3p) to undergone A-to-I editing only in the non-metastatic but not in the metastatic melanoma cells that lack ADAR1 expression[8]. A-to-I miRNAs editing can affect melanoma progression. For example, the function of miR-455-5p WT is different from its edited counterpart as they recognize different set of genes. Indeed, miR-455-5p WT but not the edited form specifically targets the tumor suppressor gene CPEB1, thus contributing to melanoma metastasis.

Here we focused our study on the relevance of miR-378a-3p editing on melanoma progression. We found that the edited form (expressed in non-metastatic cells) preferentially targets the oncogene PARVA thus preventing the progression of melanoma towards the malignant phenotype. We demonstrate that A-to-I editing in regions other than the canonical seed regions can affect miR-378a-3p binding to the 3′-UTR of PARVA. PARVA is over

expressed in metastatic melanoma cell lines and tumor specimens which have reduced expression of ADAR1. Alpha parvin, also known as actopaxin/CH-ILKB, is a member of the ILK, PINCH and parvin complex involved in the integrin-mediated signaling[10,11]. Its oncogenic roles have been previously described in breast cancer cells invasion[12] and colorectal tumor progression[13]. In addition, α-parvin promotes lung adenocarcinoma by regulating the ILK signaling pathway[14]. Regulation of ILK pathway will result in regulation of AKT and GSK3-β[14]. However, the role of PARVA in melanoma has not been described. Here, we report on a novel epigenetic mechanism regulating the expression of PARVA, and thus affecting melanoma progression.

## Results

**Identification of PARVA as a target for miR-378a-3p.** Previously we have reported that ADAR1 expression is reduced in metastatic melanoma cell lines and in clinical metastatic melanoma specimens[9]. Loss of ADAR1 directly contributes to melanoma growth and metastasis, by affecting A-to-I miRNAs editing[8]. We showed that the function of the edited miR-455-5p (expressed in non-metastatic melanoma cells) is different from its WT counterpart (expressed in metastatic melanoma cells lacking ADAR1). Indeed, WT miR-455-5p preferentially binds and inhibits the expression of the tumor suppressor gene CPEB1, thus contributing to melanoma metastasis. One of the three previously reported miRNAs undergoing A-to-I editing only in non-metastatic but not in the metastatic melanoma cells is miR-378a-3p (Supplementary Fig. 1). A-to-I editing of miR-378a-3p was observed only in the non-metastatic SB2 cells, but not in the highly metastatic C8161 cells. In addition, we extended our

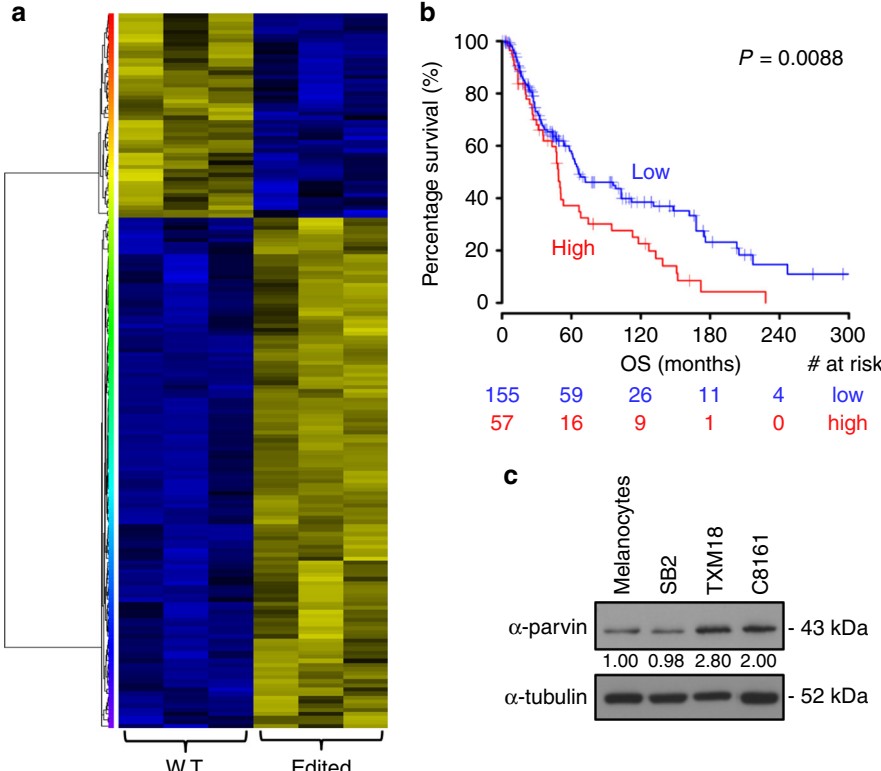

**Fig. 1** Role of PARVA in melanoma patients overall survival. **a** Microarray analysis of SB2 KD-ADAR1 cells transfected with wild type or edited miR-378a-3p. Heat map of the genes with statistically significant change expression (P < 0.01; log ratio 0.5). **b** Comparison of overall survival of skin metastatic melanoma cancer patients with high or low PARVA expression. The number of patients at risk in low/high PARVA groups at different time points are presented at the bottom of the graph. **c** Western blot analysis of melanoma cell lines shows decreased α-parvin expression in the normal melanocytes and low metastatic cells and high expression in the highly metastatic melanoma cells (TXM-18 and C8161). Representative of three independent biological samples

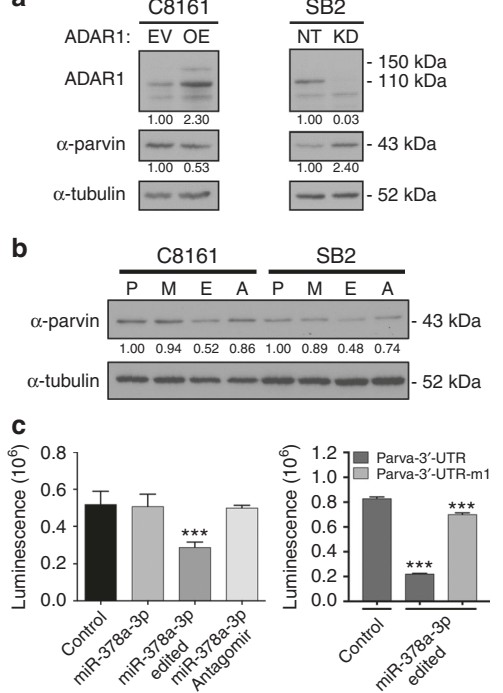

**Fig. 2** Regulation of α-parvin by miR-378a-3p. **a** Western blot analysis of C8161 and SB2 cells after overexpression or silencing of ADAR1, respectively for the expression of α-parvin. A 0.5-fold reduction in α-parvin protein expression was observed in C8161-ADAR1 overexpressing (OE) cells, whereas an increase of 2.4 fold was seen in SB2-ADAR1 knockdown (KD) cells. ADAR1 and α-parvin proteins are inversely expressed. **b** Western blot analysis demonstrating a specific regulation of α-parvin by edited miR-378a-3p (E) but not by the wild-type (M) form or its antago-miR (**A**). P represents parental cells. **a**, **b** are representative of three independent biological samples. **c** Luciferase expression driven by 3′-UTR of PARVA is decreased following expression of edited miR-378a-3p but no change was observed when wild-type miR-378a-3p was expressed (left panel). No effect was observed when the binding site at the 3′-UTR of PARVA was mutated (right panel). The data are shown as the means ± S.D., $n = 6$ (***$P < 0.001$).

editing analysis to include another pair of melanoma cell lines (A3755M vs. DX3). A-to-I editing site was observed only at the DX3 non-metastatic cells but not in the highly metastatic A375SM cell line. In addition, the overall frequency of the BLCAP mRNA, a known substrate for ADAR1 A-to-I editing, was found to be 11% in normal melanocytes but only 5.5% in metastatic melanoma cells[8]. This A-to-I editing of miR-378a-3p is not canonical and it occurs outside the seed region (Supplementary Fig. 1), and is mediated by ADAR1 and CREB expression as we previously demonstrated in the Supplementary Fig. 4 A-D of Shoshan, et al.[8]. To identify possible target genes for the WT vs the edited form of miR-378a-3p, we re-expressed either WT or the edited form of miR-378a-3p into SB-2 cells (low-metastatic, ADAR1 positive) after silencing ADAR1[8], to prevent endogenous editing. These cells were subjected to gene expression profiling. The generated heat map depicted in Fig. 1a clearly demonstrate an opposite effect mediated by the two forms of miR-378a-3p (Supplementary Data 1). Several genes have identified to be down-regulated by the edited form of miR-378a-3p (e.g., OAS1, NF2, C21orf33, IFI27, IFI6, PARVA, PARP9, YOD1, STAT1, SAMHD1). Binding prediction analysis demonstrated the predicted binding site of edited or wild-type miR-378a-3p on the 3′-

UTR of PARVA (Supplementary Fig. 2). These results places PARVA as an important player in melanoma metastasis.

We found that the oncogene PARVA was one of the gene whose expression was reduced in the presence of the A-to-I edited miR-378a-3p (Supplementary Data 1). In addition, upregulation of PARVA in ADAR1 KD cells was also reported by Nemlich et al.[9]. Considering the previous reports of the role of PARVA in other tumors we have decided to focus our study on the connection between miR-378a-3p editing and melanoma growth and metastasis. Indeed, mining the TCGA data has revealed that patients with high PARVA expression have worsen overall survival when compared with patients with low PARVA expression (Fig. 1b), indicating an oncogenic role of PARVA in melanoma. To validate the oncogenic role of PARVA in melanoma, we analyzed its expression in normal melanocytes, low-metastatic melanoma cells (SB-2) and in two highly metastatic melanoma cell lines (TXM-18 and C8161). Alpha-parvin expression is higher in metastatic melanoma cells when compared to melanocytes and early primary low-metastatic cells (Fig. 1c).

**PARVA is preferentially regulated by the edited miR-378a-3p**. To further confirm the link between ADAR1 and α-parvin expression, we overexpressed ADAR1 in C8161 (metastatic, low ADAR1 expression) and silenced its expression in SB2 cells (non-metastatic, ADAR1 positive) and tested for α-parvin expression. Figure 2a depicts an inverse correlation between ADAR1 and α-parvin expression. Silencing ADAR1 in SB2 cells resulted with a 2.4-fold increase in α-parvin expression while overexpression of ADAR1 in C8161 cells resulted with 0.5-fold reduction in α-parvin expression. To further confirm the regulation of PARVA by the edited form of miR-378a-3p, we transfected the WT and the edited forms into the C8161 and SB2 parental cells. α-parvin expression was reduced 0.5-fold in both cell lines by the edited miR-378a-3p but not by the WT or miR-378a-3p antago-miR (Fig. 2b). Similar results were confirmed at the PARVA mRNA levels (Supplementary Fig. 3). This change was not observed by other edited miRs such as miR-455-5p (Supplementary Fig. 3). We next cloned the 3′-UTR of PARVA into the pmiR luciferase reporter construct, and transfected it into miR-378a-3p WT or edited-manipulated ADAR1-KD SB-2 cells. Figure 2c shows that luciferase expression driven by the 3′-UTR of PARVA was significantly decreased by the edited form of miR-378a-3p but not when mimic WT or miR-378a-3p antago-miR were expressed, indicating that edited miR-378a-3p preferentially binds to the 3′-UTR of PARVA and inhibits its expression. Furthermore, the edited form cannot bind to the 3′-UTR of PARVA to reduce its expression when the binding site is mutated (Fig. 2c, right panel). Taken together, these data confirm that the edited miR-378a-3p can suppress PARVA expression, but the WT version cannot. Thus, the effect of the edited miR-378a-3p is mediated by its sequence-dependent ability to target PARVA.

**The role of PARVA in melanoma cell invasion**. To study the role of PARVA in melanoma progression, we first analyzed its effect on melanoma cells invasion. Silencing of PARVA in C8161 cells resulted with a significant decrease in their ability to invade through matrigel coated inserts (Fig. 3a) while over-expression of PARVA in SB2 cells resulted with increased inva-siveness (Fig. 3b). PARVA manipulation in these cells did not affect their proliferation rate (Supplementary Fig. 4). The reduction in cells invasion after PARVA silencing in C8161 cells is due to a significant decrease in MMP-2 expression (as demon-strated by western blot, Fig. 3c) and activity (as demonstrated by zymography, Fig. 3d). Silencing of PARVA in C8161 cells also

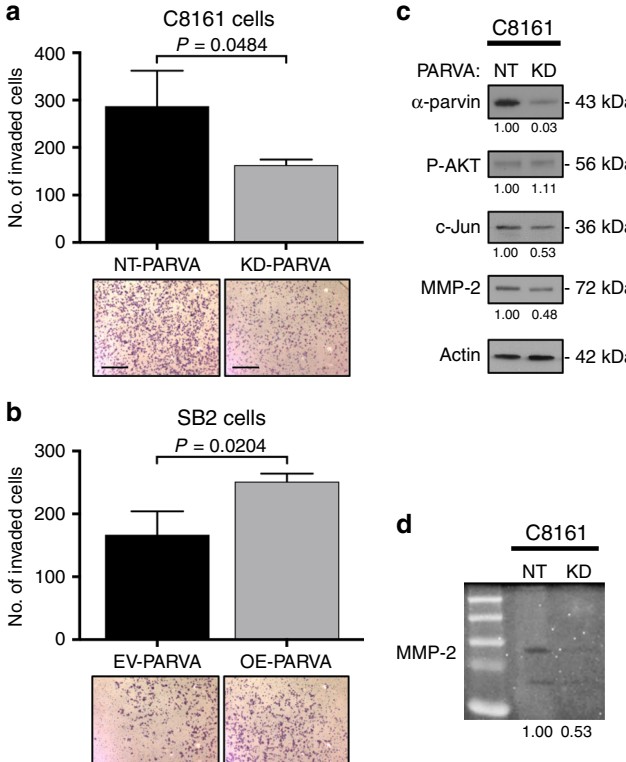

**Fig. 3** The role of *PARVA* in melanoma cells invasion. **a** C8161 KD-*PARVA* cells showed a significant reduction in the number of invaded cells (*P* = 0.0484, two-tailed Student *t* test) compared to NT-*PARVA*. Scale bars represent 200 μm. **b** SB2 overexpressing (OE)-*PARVA* cells resulted in an increased in the number of invaded cells (*P* = 0.0204, two-tailed Student *t* test) when compared to SB2 control (EV) cells. Scale bars represent 200 μm. **c** Western blot analysis of C8161 cells non-targeting (NT) or knockdown (KD) for PARVA. KD of *PARVA* in C8161 cells revealed a decreased in the protein levels of c-Jun and MMP2 but no changes in phosphorylation of AKT. **d** Zymography analysis of C8161 cells shows a reduction in the activity of MMP-2, in *PARVA* KD cells. All panels are representative of three independent biological samples

reduced c-Jun expression but did not affect AKT phosphorylation (Fig. 3c). In concordance with these findings, we found that only ADAR1-edited miR-378a-3p decreases cell invasion, but not wildtype miR-378a-3p, in both C8161 and SB2 melanoma cells (Supplementary Fig. 5).

***PARVA* contributes to melanoma growth and metastasis.** To study the role of *PARVA* in melanoma growth and metastasis in vivo, C8161 KD-*PARVA* and SB2 cells overexpressing *PARVA* were injected subcutaneously into nude mice. A significant decrease in tumor growth was observed in C8161 after *PARVA* silencing (Fig. 4a; *P* < 0.001, two-tailed Student *t* test). In contrast, overexpression of *PARVA* in SB2 cells resulted with a significant increase in tumor growth (Fig. 4b; *P* < 0.001, two-tailed Student *t* test). To study the role of *PARVA* on in vivo metastasis, C8161 KD- *PARVA* cells were injected intravenously for experimental lung metastasis assay. Silencing of *PARVA* in C8161 cells significantly reduced their ability to form lung metastases (Fig. 4c; *P* = 0.0035, two-tailed Student *t* test). Taken together, these data validate the oncogenic role of *PARVA* in melanoma. We did observe a repeated trend that treatment with the edited miRNA, reduced the number of lung metastasis although not statistically significant due to the relatively small number of animal used, which involved a complicated and highly expensive injections of

encapsulated miRNAs. The results obtained with WT miR-378a-3p is almost identical to the results when the mice were injected with the scramble control.

**miR-378a-3p contributes to melanoma metastasis.** In the last set of experiments, we wanted to elucidate the role of WT mir-378a-3p in comparison to its edited counterpart on melanoma metastasis in vivo. To that end, C8161-luciferase labelled cells were injected intravenously into nude mice. Three days later, the mice were treated 3 times weekly by intra-peritoneum injection with neutral nanoliposomes (DOPC) encapsulated with different miR-378a-3p sequences (WT-miR-378a-3p, WT-miR-378a-3p antago-miR, edited miR-378a-3p, and edited antago-miR or scramble control). Mice treated with the edited miR-378a-3p showed a significant reduction in their ability to form lung metastases when compared to mice treated with scrambled control or liposomes containing edited antago-miR at 36 days after injections (Fig. 4d).

Representative mouse from each group is shown in Fig. 4e. Collectively, these results demonstrate that the edited miR-378a-3p has an opposite effect on melanoma experimental metastasis to the WT form, and that miR-378a-3p is biologically active in melanoma.

**Discussion**
Previously we have reported that metastatic melanoma cells have reduced expression of the ADAR1 enzyme. Consequently, metastatic melanoma cells are impaired in their ability to perform A-to-I editing on microRNAs. We have identified three miRNAs: miR-455-5p, miR-324-5p and miR-378a-3p undergoing A-to-I editing only in the non-metastatic cells that express ADAR1 but not in metastatic melanoma cells lacking ADAR1 expression[8]. Indeed, re-expression of ADAR1 in metastatic melanoma cells inhibited their tumor growth and metastatic capabilities. More importantly, we previously reported that the function of the edited miR-455-5p is different from its WT counterpart[8]. For example, only the WT form of miR-455-5p binds and inhibits the expression of the tumor suppressor gene CPEB1, thus contributing to melanoma metastasis.

In the present study, we sought to determine the relevance of A-to-I editing on miR-378a-3p on melanoma progression. Unlike the case with miR-455-5p, in which we identified two edited sites within the seed sequences, the A-to-I editing in miR-378a-3p occurs at position 18, which is considered a non-canonical seed region. Nevertheless, our data support the notion that this editing event could cause a differential gene regulation when compared to its WT form, as it was recently suggested[15]. Indeed, only the edited form (found in non-metastatic cells) preferentially binds and inhibits the expression of the *PARVA* oncogene. These results demonstrate a previously unrecognized role of miRNA editing (epigenetic mechanism) in melanoma progression. In this model (depicted in Fig. 4f), we propose that activation of CREB in metastatic melanoma cells leads to downregulation of *ADAR1* expression. Subsequently, miR-378a-3p is undergoing A-to-I editing only in the early primary (non-metastatic) melanoma cells, and accumulation of the WT form in the metastatic cells lacking *ADAR1* expression. The edited form of miR-378a-3p preferentially binds and inhibits the expression of the *PARVA* oncogene, thus preventing the progression of melanoma cells towards the malignant phenotype. It should be noted, however, that the *PARVA* mRNA is not the only target for the edited form of miR-378a-3p. Indeed, we have identified several target genes that their regulations are influenced by miRs editing contributing to melanoma growth and metastasis. This concept is supported by our in vivo studies, demonstrating that a delivery of the edited

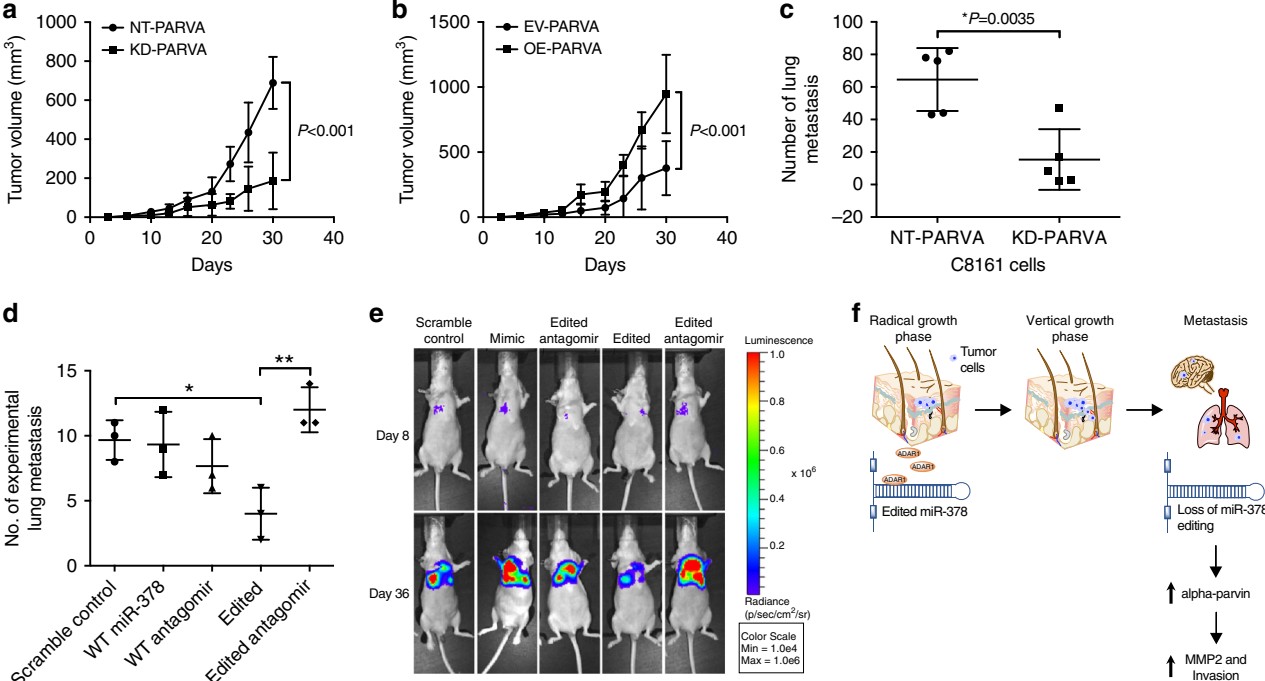

**Fig. 4** Contribution of *PARVA* and miR-378a-3p to melanoma tumor growth and metastasis in vivo. **a, b** Effect of manipulation of *PARVA* on subcutaneous tumor growth. **a** Knockdown (KD) of *PARVA* in C8161 cells ($1 \times 10^6$) resulted with a significant reduction in tumor growth compared with the non-targeting (NT) control, $P < 0.001$. For each group, $n = 5$ mice, statistical significance by two-tailed Student $t$ test, error bars represent s.d. **b** Overexpression (OE) of *PARVA* in SB2 cells ($1 \times 10^6$) significantly increased tumor growth compared to the empty vector (EV) control, $P < 0.001$. For each group $n = 5$ mice, statistical significance by two-tailed Student $t$ test, error bars represent s.d. **c** Silencing of *PARVA* in C8161 cells significantly reduced the number of experimental lung metastases, $P = 0.0035$. Statistical significance by two-tailed Student $t$ test; error bars represent s.d. **d, e** C8161 luciferase labelled cells ($2 \times 10^5$) were injected intravenously, and 3 days later the mice were treated via intra-peritoneum injection with either scramble control, miR-378a-3p wild-type (WT), WT antago-miR, edited miR-378, or edited antago-miR encapsulated in 1, 2-dioleoyl-sn-glycero-3-phosphocoline (DOPC) particles. The data shown represent 36 days after injections. **d** Delivery of edited miR-378a-3p to mice harboring C8161 cells ($2 \times 10^5$) decreased their metastatic potential when compared to scramble control ($P \le 0.05$) and when compared to its antago-miR ($P \le 0.01$). Statistical significance by Tukey's multiple comparisons test. For each group $n = 3$. **e** Representative luciferase imaging at 36 days after cells injection. **f** Model of miR-378a-3p editing in melanoma progression. *ADAR1* is lost in the transition from radial growth phase to vertical growth phase, consequently there is an accumulation of WT miR-378a-3p in metastatic melanoma causing overexpression of PARVA and MMP-2 thus contributing to melanoma invasion and metastasis

but not the WT form of miR-378a-3p by means of nanoparticles inhibited the metastatic capabilities of melanoma cells. Other studies have reported that miR-378a-3p plays a role in diverse cancers[16–19]. Hyun et al. showed that miR-378a-3p reduced the activation of hepatic stellate cells by binding to the 3′-UTR of Gli3[20], while in lung adenocarcinoma, miR-378a-3p reversed the chemo-resistance to cisplatin by targeting clusterin[17]. These studies, however, did not distinguish between the edited and non-edited forms of miR-378a-3p.

Previous publications have shown that *PARVA* plays a role in the degradation of extracellular matrix leading to cell invasion via regulation of Rho GTPase signaling in breast cancer[21], and in hepatocellular carcinoma (HCC) progression and metastasis[22]. The current study identified the *PARVA* gene to be a direct target for miR-378a-3p (its edited form) and outlined the contribution of *PARVA* to melanoma growth and metastasis. Overall, these results provide a key missing link in the mechanistic epigenetic pathway of A-to-I miRNAs editing in the acquisition of the melanoma metastatic phenotype. At this stage, further clinical studies to determine the relevance of micro-RNA editing in human melanoma progression must be conducted.

## Methods

**Cell culture**. Human melanoma cell lines were culture in MEM media supplemented with 10% fetal bovine serum, 2 mM glutamine, 1% non-essential amino acids, and 1% antibiotic antimycotic. 293T cells (Invitrogen) were used to generate lentivirus for overexpression or knockdown of the genes of interest. These cells were grown in DMEM supplemented with 10% FBS. Primary normal human melanocytes cells were obtained from Promo Cell (Cat. # C-12403). The following human melanoma cells were used: SB2 cells poorly metastatic that was originated from early primary cutaneous melanoma patient[23], C8161 and TXM18 are highly metastatic melanoma cell lines that were originated from patient's lymph nodes[23,24]. All cell lines used in our studies were established at the University of Texas M. D. Anderson Cancer Center and tested before their usage for authentication by DNA fingerprinting using the short tandem repeat method. Cell lines were routinely tested to confirm the absence of Mycoplasma.

**Microarray**. SB2 KD-*ADAR1* cells transfected using RNA iMAX transfection reagent in Opti-MEM I Gluta MAX reduced serum media with either wild-type or edited miR-378a-3p, in triplicates. Cells were incubated overnight and fresh media was added. After 24 h, the media was removed and RNA was purified using RNA mini-prep (Zymo Research). Gene Chip Human Genome U133 plus 2.0 array analysis was performed. Gene expression profile was determined using Nexus and uploaded to the Gene Expression Omnibus (GEO) repository with the accession number GSE107088.

**Western blot analysis**. Protein samples from human melanoma cells were subjected to SDS-PAGE and Western Blot analysis. Briefly, cells were lysed using RIPA B buffer containing proteases and phosphatases inhibitors (ROCHE) to extract the proteins. Protein concentration was determined using Bio-Rad protein assay (Bio-Rad) and the Epoch spectrophotometer (BioTek). SDS-poly-acrylamide gels were used to separate the proteins by molecular weight. Western blots were performed to detect the following proteins of interest: ADAR1 (Sigma, Cat. No. SAB4200541), α-tubulin (Sigma, Cat. T-5168), actin (Sigma, Cat. A-20665168), c-Jun Ser 63/73 (Santa Cruz, Cat. No. sc-16312), α-parvin (Cell Signaling, Cat. No. 4026), AKT Ser 473 (Cell Signaling, Cat. No. 9271) and MMP2 (Cell Signaling, Cat. No. 4022). Supplementary Fig. 6 shows the full blots.

**Invasion assay**. Bio-coat growth factor reduced, matrigel invasion chambers with 8.0 μm membrane (Corning) were used to perform cell invasion assay. Either 20,000 (C8161) or 50,000 (SB2) cells were plated on the top chamber in serum-free media. The bottom chamber contained complete media supplemented with 10% FBS. After 16 h, the top cells were wiped out with a cotton tip and the membranes were stain with HEMA 3 (Fisher Scientific). Membranes were cut from the chamber and mounted on slides. The numbers of invaded cells were counted under the microscope. Triplicates of each condition were performed.

**MTT proliferation assay**. Cells were plated on 96-wells plates at concentrations of 2000 or 5000 cells per well. MTT assay were performed for 5 consecutive days. MTT was dissolved at a concentration of 5 μg/mL in sterile PBS. To each well, 20 μL of MTT were added and incubated for 2 h. After 2 h media was removed and 200 μL of DMSO was added into each well. O.D. was measured using Epoch (Bio Tek) spectrophotometer. Each sample has 12 replicas.

**Luciferase assay**. SB2 KD-*ADAR1* cells were infected with Mission *PARVA* 3′-UTR Lenti GoClone, Mission *PARVA* 3′-UTR-mutated Lenti GoClone or non-target control1 (Sigma) and selected with puromycin. Micro-RNAs against WT miR-378a-3p, edited miR-378a-3p or miR-378a-3p antago-miR (Ambion) were transfected using RNA iMAX transfection reagent (Invitrogen). Luciferase Assay (Promega) was used to determine the firefly luciferase activity and each sample was normalized with its non-targeting control. Each sample has six biological replicas.

**Plasmids and constructs**. *PARVA* was cloned from A375 melanoma cells cDNA into pCDH-CMV-MCS-EF1-puro expression vector (CD510B-1, SBI) with the following primers. *PARVA*-XbaI-F: 5′-GCTCTAGAATGGCCACCTCCCCGCA-GAAGTC-3′ and *PARVA*-NotI-R: 5′-ATAAGAATGCGGCCGCTCACTCC ACGTTACGGTACTTGG-3′. All constructs were fully sequenced in both direction.

For the *PARVA* mutant luciferase analysis, the two binding site for miR-378a-3 promoter region were deleted: 5′-CCCAAGAGTTCTTGCTGTTGGCGTACTGG ACCCTCCTCCGAACTGCCTTACCC TGCTTATTCCTGTCTCTTGCACT **GTGCTCTCCCACAAGT CCAGCTGCAACCCAGAGATAGTGGAAA CTGAAAT**TAGGAAGGAAATCATCAATAA CTCAGTGGGCTGACCCATCC CTCCCAGGCGCTGGGGACCAACCTAGCAATGAAGGTTGGGAAGGT TGTTCCCTTCCCGG**TGCCAGGTCCAGATTTCC CTCCATGATTTGG GAACCAGC TTAGGCAAAAG**AGTCCCCACAAGATGAAAATAAAGATC CTAGTTACCAT TCAAAGGATGCT-3′ (highlighted "bold" area shows the binding region with the deleted region in grey font). Light-grey fonts indicate the deleted region. The fragment was digested with HindIII and SepI and ligated into the pGL3-basic vector (Promega), digested with KpnI and XhoI and ligated into the pGL3-basic vector (Promega). Site-directed mutagenesis of the PARVA sites (deletion), was carried out using the QuikChange II XL site-directed mutagenesis kit (Stratagene) according to the manufacturer's instructions.

**In vivo experiments**. SB-2 cells were stably transduced with Lentivirus-based *PARVA* expression or empty vector and with pGreenFire positive control vector containing GFP and Luciferase (TR011PA-1. SBI). After transduction, cells were selected by puromicin and sorted after 10 days by flow cytometry using GFP expression. C8161 cells were stably transfected with shRNA for *PARVA* KD. *PARVA* expression levels were subjected to western blot analyses to verify protein overexpression or KD.

Athymic female nude mice Nu/Nu with an age range of 8–12 weeks were used to determine the in vivo effect of *PARVA* and miR-378a-3p editing. First, the in vivo effect of *PARVA* regulation was studied. $1 \times 10^6$ of C8161 NT-control or KD *PARVA* and SB2 EV-control or OE-*PARVA* were subcutaneously injected. Tumor volume was recorded three times a week using a caliper. Tumor volume was determined by (wide$^2$) × (length) / 2.

The metastatic potential of the *PARVA* manipulated cells was determined using $1.0 \times 10^6$ C8161 NT-control or KD-*PARVA* cells by tail vein intra-venous (i.v.) injection. After i.v., we waited 3 days before starting treatment with encapsulated miRs including WT, edited or antagomir. Mice were monitored daily for 36 days. Lungs were collected, fixed in Bouin fixative (Ricca Chemicals) and lung colonies were counted.

The miRNA oligos were incorporated into 1,2-dioleoyl-sn-glycero-3-phosphocholine (DOPC; Avanti Polar Lipids, Inc.) by lyophilization. Briefly, 5 μg RNA oligo and DOPC were mixed in the presence of excess t-butanol at a ratio of 1:10 (w/w) as described previously[25]. After addition of Tween-20, the mixture was frozen in an acetone-dry ice bath and dried. Liposomes were reconstituted by adding 200 μL water into the vial and mixing briefly.

DOPC liposomes (containing control, miR-378a-3p, antago-miR, edited miR-378a-3p or edited antago-miR) were injected via intra-peritoneum three times a week, starting 3 days after i.v. tail vein injections of $2.0 \times 10^5$ luciferase labelled C8161 cells.

In vivo imaging was performed to monitor mice lung tumor burden. Mice were anesthetized using 2% inhaled isofluorane. Bioluminescence imaging were obtained, using IVIS 100 series system (Xenogen), 12 min after i.p. injection of 0.2 mL D-luciferin (15 mg/mL, Gold Bio Technology). After killing, lungs were fixed in Bouin solution and the number of lung colonies were counted using a dissecting microscope (Nikon). Statistical significances were analyzed by the Tukey's multiple comparisons test. All animal experiments were conducted in accordance with American Association for Laboratory Animal Science regulations and the approval of The University of Texas MD Anderson Cancer Center Institutional Animal Care & Use Committee.

**TCGA analysis**. The mRNA expression of *PARVA* for the TCGA Skin Cutaneous Melanoma cohort is the RNASeqv2 level 3 data sets downloaded from Broad GDAC Firehose http://gdac.broadinstitute.org/. The clinical information for this cohort was obtained from cBioPortal (http://www.cbioportal.org).

Survival analysis was performed in R (version 3.2.5). The relationship between overall survival and covariates (mRNA expression levels and clinical parameters, age at diagnosis, gender and stage as there was available information for most of the patients) was examined using a Cox proportional hazard model. Age at diagnosis, stage and *PARVA* mRNA were significant in the univariate analysis and a multivariate Cox proportional hazard model was fitted to this data. *PARVA* expression in metastatic melanomas is an independent poor prognostic marker.

In order to visualize survival differences between groups of patients based on *PARVA* levels, we used the log-rank test to find the point (cut-off) with the most significant (lowest *P* value) split into high and low mRNA level groups. The Kaplan-Meier plots were generated for these cutoffs. The median survival time of each group is presented in brackets. The numbers of patients at risk in low and high mRNA groups at different time points are presented at the bottom of the graph. For overall survival, the cut-off to optimally separate the patients in high, low (min *P* value) group was 0.63.

Supplementary Table 1 and 2 show *PARVA* expression in metastatic melanomas and clinical information for the cohort.

**Data availability**. The authors declare that the data supporting the findings of this study are available within the paper and its supplementary information files. Any other relevant data that support the findings of this study are available from the corresponding author upon reasonable request. The normalized gene expression profile data generated from Gene Chip Human Genome U133 plus 2.0 array analysis supporting the opposite effect mediated by the two forms of miR-378a-3p (Fig. 1 and Supplementary Data 1) has been deposited in the Gene Expression Omnibus (GEO) repository with the accession code GSE107088 [https://www.ncbi.nlm.nih.gov/geo/query/acc.cgi?acc=GSE107088].

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

## Acknowledgements

We thank the Animal Resources Technology Department of Veterinary Medicine and Surgery at UTMDACC. This study was support by SINF, MDACC grant, and NIH Skin Cancer SPORE P50 CA 093459. Dr. Fuentes-Mattei was supported in part by Award Number P50 CA140388 from the NCI and by the NIH Clinical Research Loan Repayment Program. Dr. Calin was supported in part by the National Institutes of Health (NIH/NCATS) grant UH3TR00943, NIH/NCI R01-CA182905, the UT MD Anderson Cancer Center SPORE in Melanoma grant from NIH/NCI P50 CA093459, AIM at Melanoma Foundation, the Miriam and Jim Mulva research funds, and a CLL Moonshot Flagship project.

## Author contributions

Study concept and design: G.V-T. and M.B-E. Acquisition of data: G.V-T., E.S., C.I., L.H., E.F-M., H.P., S.J.K., C.R-A., and V.X. Provided the DOPC nanoparticles encapsulated with miRs: G.L-B. and A.S. Analysis and interpretation of data: G.V-T., C.I., E.F-M., D.B., S.J.M.J., A.G.R., G.C. and M.B-E. Drafting of the manuscript: G.V-T. and M.B-E. Critical revision of the manuscript: G.V-T., E.S., C.I., E.F-M., A.G.R. and M.B-E. Funding of the study: M.B-E.

## Additional information

**Competing Interests:** The authors declare no competing financial interest.

