## [Peer Review File · Nature Communications]

Reviewers' comments:

Reviewer #1 (Remarks to the Author):

To the authors,

Valazquez-Torres et al., extend on their previous publication of how ADAR1 A-I editing can alter metastasis in melanoma. The authors describe how an edited miRNA can reduce the expression of a-parvin (PARVA) and correlate this to tumour invasion. miR-373a-3p was initially described by the group in a screen of invasive and non-invasive edited miR.

At this time the data does not represent the same coherent story as the authors would like to argue.

[Please do not confuse my efficient style and brief wording as an indication of my mood].

1) Supplementary Figure 1:

- How was this data obtained?
- Is this the only editing of miR-373a-3p?
- what is the frequency of this editing within the cells (i.e. % edited vs non-edited)?

2) Line 146 should read "(Supplemental 2)"

3) Supplemental Figure 2A should only be "2" as there are no extra parts

4) Figure 1B

- TCGA database should be properly referenced
- how was the data-mining achieved

5) Figure 1A - why was SB2 KD-ADAR1 cells used instead of C8161?

- needs to be addressed in the text

6) The journal has no length restrictions; the current Methods section can only be understood in conjunction with previous (not indicated) publications. More detail is needed (e.g. cell line origins, antibody cat numbers and possible lot numbers, how in the in vivo experiments were the miR encapsulated). The current work cannot be reproduced with the Methods described.

7) Critical: What are the time lines of the experiments. e.g. After transfections how long are the authors waiting to until they analyse the effect. After injections etc.

8) Figure 2A

- does OE of ADAR1 lead to enhanced miR-373a-3p editing?
- 9) Figure 2C, because the miR-373a-3p does not bind the seed region
- does miR-373a-3p lead to a loss of PARVA mRNA?
- do other ADAR1 edited miR also alter a-parvin levels?

10) Figure 3

- what is the effect of miR-373a-3p on invasion?
- KD of PARVA is >10x more potent than miR-373a-3p editing knock-down
- why is actin instead of a-tubulin used here?

11) Figure 4

- 4C vs 4D - the untreated control (NT-PARVA in 4C) has 45 - 80 metastasis yet the scrambled control (theoretically identical to NT-PARVA) has only 7 - 11? Explain please. Does this mean adding random miR reduces metastasis by 7 to 10 fold?
- Since there is no significant difference between edited and unedited miR-373a-3p, please discuss and speculate.
- Model in 4F is not needed

General:

- Given that ADAR1 has a profound effect on tumour metastasis and can edited a variety of miRNA, it is not clear that the presented data is showing a direct connection between ADAR1 and miR-373a-3p editing.
- Given that miR can alter a variety of targets, the edited miR-373a-3p could also influences a number of other targets which have not been tested in the invasion or metastasis model. Given

that a-pavrin was not a top 10 hit all in vivo effects of miR-373a-3p are to be analysed with this in mind.

- The authors failed to discuss the function of a-pavrin in metastasis from previous publications (2012 and 2013). The effects in those publications were more dramatic than here, hence Figure 3 and 4 have lost their impact.

Reviewer #2 (Remarks to the Author):

In this manuscript, Velazquez-Torres et al extend previous finding in which they have established the impact of reduced ADAR-1 expression in A-to-I micro-RNA editing. Here, authors provide evidence suggesting that miR-378a-3p is subject to this editing, which takes place only in the non-metastatic melanoma cell lines. Authors further demonstrate that the edited form of miR378a-3p preferentially binds to the 3'UTR of the PRAVA oncogene and inhibits its expression, which attenuates melanoma progression. The following would be helpful to clarify and substantiate authors conclusions.

1. What makes PRAVA selection? Are there other genes that would be subjected to similar regulation by the edited miRNA?
2. What confer the selection of the editing to the miRNA studied here?
3. Would melanoma in which the 3'UTR of PRAVA (natural occurring SNPs or CRISPRed lines) no longer obey the regulation by miR-378a-3p?
4. How wide is the phenomenon studied here? How many melanomas exhibit altered editing where both miR-378a and PRAVA expression are respectively altered and degree of metastasis is correspondingly changed?

Additional points

Figure 1 – data for melanomas where endogenous levels of MiR378a—3p are altered need to be provided. Corresponding changes for PRAVE in these lines should be evaluated.

Figure 2- the differences seen are not very compelling. Are there natural occurring melanoma with altered expression of components reported to be part of the regulatory axis studied here?

Figure 3 – the differences shown are not very compelling and westerns are largely weak. Changes in PRAVA-dependent invasion of melanoma should be demonstrated via altered miR 378a-3p.

Figure 4D-E the changes observed upon administration of the miR378-a-3p to melanoma should be accompanied by co-manipulation of PRAVA to establish the causative role of PRAVA in the effects attributed for miR378a-3p.

REVIEWER #1:

1) Supplementary Figure 1:

- **How was this data obtained?**
- **Is this the only editing of miR-378a-3p?**
- **What is the frequency of this editing within the cells (i.e. % edited vs non-edited)?**

Response:

- This data was obtained by subjecting the cell lines to deep RNA sequencing analysis. This is now described in the Figure's Legend, with a reference to our previous Nat. Cell. Biol. paper.

- This is the only A-to-I editing site observed for miR-378a-3p

- We extended our editing analysis to include another pair of melanoma cell lines (A3755M vs. DX3). A-to-I editing site was observed only at the DX3 non-metastatic cells but not in the highly metastatic A375SM cell line. In addition, the overall frequency of the BLCAP mRNA, a known substrate for ADAR1 A-to-I editing, was found to be 11% in normal melanocytes but only 5.5% in metastatic melanoma cells. Please see Figure 1h in our NCB paper.

2) Line 146 should read "(Supplemental 2)"

Response:

- This line is now corrected to read: (Supplementary Figure 1). Now, it appears in page 9, line 210.

3) Supplemental Figure 2A should only be "2" as there are no extra parts

Response:

- Supplementary Figure 2 is now only 2.

4) Figure 1B

- **TCGA database should be properly referenced**
- **how was the data-mining achieved?**

Response:

- TCGA analysis is now described in details and properly referenced in the Methods section. Please see page 8, lines 175-194.

- The mining of the data is also described in this section, including the survival data shown in Figure 1B.

5) Figure 1A - why was SB2 KD-ADAR1 cells used instead of C8161?

- **needs to be addressed in the text**

Response:

- The logic behind this approach was to use cells without any endogenous expression of ADAR1 so that the WT or the introduced edited forms will not be edited by the endogenous ADAR1. It is described in the text (Results section page 9, lines 210-213)

6) The journal has no length restrictions; the current Methods section can only be understood in conjunction with previous (not indicated) publications. More detail is needed (e.g. cell line origins, antibody cat numbers and possible lot numbers, how in the in vivo experiments were the miR encapsulated). The current work cannot be reproduced with the Methods described.

Response:

- We now added to the methods section all of the missing information, including cell line origins (page 4, lines 75-78), antibody cat numbers, how the in vivo experiments were performed including encapsulation procedures with previous references (page 7, lines 144-167).

7) **Critical: What are the time lines of the experiments. e.g. After transfections, how long are the authors waiting to until they analyze the effect. After injections, etc.**

Response:

- We now added a detailed description of our in vivo experiments. Briefly, SB-2 cells were stably transfected with Lentiviruses-based GFP-labeled expression vector for PARVA and sorted after 10 days by flow cytometry. C8161 cells were stably transfected with shRNA for PARVA KD and sorted by puromycin selection. Then, PARVA expression levels were subjected to western blot analyses to verify overexpression or KD. The cells were injected s.c. for tumor growth or i.v. for experimental lung metastasis assay. In the case of the i.v. injections we waited 3 days (shown in Fig 4C and E) before starting treatment with encapsulated miRs including WT, edited and antagomir. The results depicted in Figures 4D and E represent the end point of 36 days. This description is now added to the Method section (page 7, line 158), to Results section (page 12, line 291) and to Figure Legend (page 24, lines 535).

8) **Figure 2A**

- **does OE of ADAR1 lead to enhanced miR-373a-3p editing?**

Response:

- The answer is yes. We overexpressed ADAR1 in C8161 cells, which resulted with enhanced miR-378a-3p editing. This data is shown in our NCB paper, supplemental Figure 4 A-D.

9) **Figure 2C, because the miR-373a-3p does not bind the seed region**

- **does miR-373a-3p lead to a loss of PARVA mRNA?**

- **do other ADAR1-edited miR also alter α -parvin levels?**

Response:

- As suggested by the Reviewer, we performed a new set of experiments to demonstrate the effect of edited miR-378a-3p on PARVA mRNA levels in melanoma cells. We found that the only edited form of miR-378a-3p downregulated PARVA mRNA. Please see the new supplemental Figure 3 depicting the results.

-Other edited miRs such as miR-455-5p did not affect PARVA levels. These results are now shown in the newly added supplemental Figure 3.

10) Figure 3

- **what is the effect of miR-373a-3p on invasion?**
- **KD of PARVA is > IOx more potent than miR-373a-3p editing knock-down**
- **why is actin instead of a-tubulin used here?**

Response:

- In response to the Reviewer's suggestion, we examined the effect of WT vs. the edited form of miR-378a-3p on cell invasion in melanoma. The results clearly show that edited miR-378a-3p but not the WT form, reduces cell invasion of C8161 melanoma cells through matrigel invasion assay. These results are now presented in the newly Supplementary Figure 5.

- It is expected that direct KD of PARVA will have a much potent effect on PARVA expression as compared with the edited form of miR-378a-3p which is not the only factor affecting PARVA expression.

- The Reviewer is right raising this question. Indeed, we usually use tubulin as a reference for the loading control expect when the MW is too close and interfere or masking one of our bands, especially dealing with phospho AKT in this particular gel.

11) Figure 4

- **4C vs 4D - the untreated control (NT-PARVA in 4C) has 45 - 80 metastasis yet the scrambled control (theoretically identical to NT-PARVA) has only 7 - 11? Explain please. Does this mean adding random miR reduces metastasis by 7 to 10 fold?**
- **Since there is no significant difference between edited and unedited miR-373a-3p, please discuss and speculate.**
- **Model in 4F is not needed**

Response:

-The explanation for this discrepancy is simply due to performing two different experiments i.e. in 4C, we injected 1×10^6 cells while in the experiments described in Figure 4D, we injected only 2×10^5 cells, hence the difference in the number of lung metastasis. This is now explained and added in the methods section as well as the figure's legend. Based on this experiments and previous other experiments performed in the lab, adding random miR, dose not reduce metastasis by 7 to 10 fold.

- We did observe a repeated trend that treatment with the edited miR, reduced the number of lung metastasis although not statistically significant due to the relatively small number of animal used which involved a complicated and highly expensive injections of encapsulated miRs. The results obtained with WT miR-378a-3p is almost identical to the results when the mice were injected with the scramble control.

- We strongly believe that the model depicted in Figure 4F is a good summary for our working model on how the edited miR-378a-3p affect melanoma growth and metastasis, and it simplifies the overall presented data for the readers.

General:

- **Given that ADAR1 has a profound effect on tumor metastasis and can edited a variety of miRNA, it is not clear that the presented data is showing a direct connection between ADAR1 and miR-373a-3p editing.**

Response:

Indeed, ADAR1 can edit a variety of miRs as we previously reported. In the present paper, we decided to concentrate on ADAR1 and miR-378a-3p editing. Re-expression of ADAR1 in metastatic melanoma cells was sufficient and enough to inhibit their tumor growth and their metastatic capabilities. These points raised by the Reviewer are now discussed in the Discussion section in pages 12-13, lines 297-306.

- **Given that miR can alter a variety of targets, the edited miR-373a-3p could also influences a number of other targets which have not been tested in the invasion or metastasis model. Given that a-parvin was not a top 10 hit all in vivo effects of miR-373a-3p are to be analyzed with this in mind.**

Response:

The Reviewer correctly pointed to the argument that PARVA is not the only gene regulated by the edited miR. Indeed, we have identified several other target gene regulated preferentially either by the WT or the edited form of miR-378a-3p. The point that the edited miR-378a-3p could also influence a number of other genes contributing to melanoma growth and metastasis is now added to the Discussion section page 9, lines 215-217.

- **The authors failed to discuss the function of a-parvin in metastasis from previous publications (2012 and 2013). The effects in those publications were more dramatic than here, hence Figure 3 and 4 have lost their impact.**

Response:

The function of α -parvin in metastasis from previous publications mentioned by the Reviewer are now cited and discussed. (page 14, lines 331-333, references 24 and 25). While the role of α -parvin (PARVA) in metastasis has been described previously for other tumors, Figure 3 and 4 demonstrate, for the first time, the role of PARVA in melanoma and therefore their impact on on the field of melanoma metastasis is crucial and has to be demonstrated.

REVIEWER #2

- 1. What makes PARVA selection? Are there other genes that would be subjected to similar regulation by the edited miRNA?**

Response:

We have identified several other genes that would be subjected to similar regulation by the edited miR-378a-3p (please see supplementary Table 1 and the heat map depicted in Fig 1A). The reason why we decided to concentrate on the connection between miR-378a-3p and PARVA is now described in the Result section (page 9, lines 215-221). Please also see our answer to Reviewer 1, who raised the same issue. We also now emphasize (see discussion on page 13, lines 321-327) that PARVA is not the sole target gene and potentially there might be other genes regulated by the

edited form contributing to melanoma growth and metastasis. Given the previous publication on the role of PARVA on metastasis on other tumors was among the reasons we selected to concentrate on PARVA (Described now in the result section page 10, lines 224-226).

2. What confer the selection of the editing to the miRNA studied here?

Response:

As we reported previously, we have identified 3 miRNAs (miR455-5p, miR-324-5p and miR-378a-3), to be undergoing A-to-I editing only in the non-metastatic melanoma cells expressing ADAR1. The identification of these miRs and their sites of editing is described in our NCB paper and referenced in the current manuscript.

3. Would melanoma in which the 3'UTR of PRAVA (natural occurring SNPs or CRISPRed lines) no longer obey the regulation by miR-378a-3p?

Response:

To answer the Reviewer question, we have performed and added a new Figure (Figure 2C, right panel) in which we cloned the 3'UTR of PARVA in front of a Luciferase reporter gene. Utilizing these constructs, we were able to demonstrate that the edited form could not bind to 3'UTR of PARVA when the binding site was mutated and thus no longer obey the regulation by miR-378a-3p. This is now depicted in the newly formatted Figure 2C, including the Figure's legend. The constructs and their generation are now described in the Methods section on page 6, lines 123-142.

4. How wide is the phenomenon studied here? How many melanomas exhibit altered editing where both miR-378a and PARVA expression are respectively altered and degree of metastasis is correspondingly changed?

Response:

We joined forces with Dr. Gordon Robertson (Vancouver, Canada, co-author on this manuscript) in an attempt to mine the TCGA data for the frequency of miR editing in melanoma. Our analysis was inconclusive due to the low number of early primary melanomas deposited into the TCGA bank. We started to collect our own specimens at M.D. Anderson. Once we collect enough early primary samples we will subject them to RNA deep sequencing analysis. Mining the TCGA data did reveal however, a direct correlation between PARVA expression and melanoma patients' survival. It is for this reason and to be objective we added the last sentence in the discussion to read: "At this stage, further clinical studies to determine the relevance of micro-RNA editing in human melanoma progression must be conducted".

Additional points:

Figure 1 - Data for melanomas where endogenous levels of miR-378a-3p are altered need to be provided. Corresponding changes for PARVA in these lines should be evaluated.

Response:

These experiments were performed and presented in Figure 1C demonstrating a direct correlation between PARVA expression and melanoma metastasis. The two highly metastatic melanoma cell lines TXM18 and C8161 express higher levels of PARVA when compared to normal melanocytes and the non-metastatic SB2 (both of which express ADAR1 and capable to perform A-to-I RNA

editing). In addition, the Kaplan-Meyer plot depicted in Figure 1B clearly shows a direct correlation between PARVA expression and patient survival. To directly answer the Reviewer question, the levels of the endogenous miR-378a-3p in C8161 and SB2 were manipulated to express either mimic antagomir or the edited form of miR-378a-3p (Figure 2C). The levels of the manipulated miRs were verified by RT-PCR and ensued by analyzing the levels of PARVA. Figure 2C now show that only the expression of the edited form affected the levels of α -parvin expression.

Figure 2- the differences seen are not very compelling. Are there natural occurring melanoma with altered expression of components reported to be part of the regulatory axis studied here?

Response:

Naturally occurring melanoma with altered expression of the axis ADAR1,miR-editing and PARVA are shown in Figure 1C in which the two highly metastatic melanoma cell lines TXM18 and C8161 do not expressed ADAR1, are negative for miR-378a-3p editing and express high levels of α -parvin. This is validated in real life in patients by the Kaplan-Meyer curve presented in Figure 1B.

Figure 3 - the differences shown are not very compelling and westerns are largely weak. Changes in PRAVA-dependent invasion of melanoma should be demonstrated via altered miR-378a-3p.

Response:

As requested by the Reviewer, the effect of WT vs the edited form of miR-378a-3p on melanoma invasion was examined. The results are depicted in the newly added Supplementary Figure 5 which clearly show that introduction of the edited form of miR-378a-3p into C8161 cells reduced their invasive potential (see also Results section, page 11, lines 258-269).

Figure 4D-E the changes observed upon administration of the miR378-a-3p to melanoma should be accompanied by co-manipulation of PRAVA to establish the causative role of PRAVA in the effects attributed for miR378a-3p.

Response:

In these in vivo experiments, we sought to concentrate on the effect of miRs manipulation on melanoma metastasis. Co-manipulation of PARVA in these experiments is too complicated to perform along with DOPC-encapsulated miRs. In addition, since the role of PARVA in tumor invasion and metastasis is very well established we felt that co-manipulation to establish the causative role of PARVA is redundant. It was more important for us to show the differential effect of WT vs the edited form of miR-378a-3p on PARVA expression.

REVIEWERS' COMMENTS:

Reviewer #1 (Remarks to the Author):

The authors have addressed the majority of the previously described concerns.

The following aspects were not addressed satisfactorily:

For 1) **What is the frequency of this editing within the cells (i.e. % edited vs non-edited)?**

The authors should have presented the data on how many times the edited miR-378a-3p was detected as compared to the unedited miR-378a-3p. This should be included in the text.

For 11) **Since there is no significant difference between edited and unedited miR-373a-3p, please discuss and speculate.**

Although the authors provided the reviewer with an explanation, the reader might still want to ask the same. The authors' explanation should be added to the text.

General:

The authors provide good explanations for the raised discrepancies. Overall, this reviewer finds that the possibilities of indirect or secondary effects warrant a softening of the claims.

The title should be changed from

A-to-I miR-378a-3p editing prevents melanoma progression via regulation of PARVA expression to

A-to-I miR-378a-3p editing **can prevent** melanoma progression via regulation of PARVA expression

Reviewer #2 (Remarks to the Author):

Authors have largely addressed the reviewer comments and the manuscript has thus been improved.

RESPONSE TO REVIEWERS

REVIEWER #1:

Reviewer #1 has suggested to change the title to: A-to-I miR-378a-3p editing can prevent melanoma progression via regulation of PARVA expression.

Response:

- We accept his suggestion and changed the title accordingly.

Reviewer #1 also raised two other minor points:

- 1) For 1- What is the frequency of this editing within the cells (i.e. % edited vs non-edited)?**

The authors should have presented the data on how many times the edited miR-378a-3p was detected as compared to the unedited miR-378a-3p. This should be included in the text.

Response:

We now added our response to the Reviewer in the text. Please see page 4 lines 80-86.

- 2) For 11- Since there is no significant difference between edited and unedited miR-373a-3p, please discuss and speculate.**

Although the authors provided the reviewer with an explanation, the reader might left to ask the same. The authors explanation should be added to the text.

Response:

As suggested by the Reviewer, the explanation is now added to the text. Please see page 7 lines 164-169.

REVIEWER #2:

Reviewer #2 stated that the Authors had largely addressed all of the previous comments with no further requests.